# Delays in Epidemic Outbreak Control Cost Disproportionately Large Treatment Footprints to Offset

**DOI:** 10.3390/pathogens11040393

**Published:** 2022-03-24

**Authors:** Paul M. Severns, Christopher C. Mundt

**Affiliations:** 1Department of Plant Pathology, University of Georgia, Athens, GA 30602, USA; 2Department of Botany and Plant Pathology, Oregon State University, Corvallis, OR 97331, USA; mundtc@science.oregonstate.edu

**Keywords:** outbreak control, disease management, epidemic control, ring cull, quarantine area, rapid outbreak response

## Abstract

Epidemic outbreak control often involves a spatially explicit treatment area (quarantine, inoculation, ring cull) that covers the outbreak area and adjacent regions where hosts are thought to be latently infected. Emphasis on space however neglects the influence of treatment timing on outbreak control. We conducted field and in silico experiments with wheat stripe rust (WSR), a long-distance dispersed plant disease, to understand interactions between treatment timing and area interact to suppress an outbreak. Full-factorial field experiments with three different ring culls (outbreak area only to a 25-fold increase in treatment area) at three different disease control timings (1.125, 1.25, and 1.5 latent periods after initial disease expression) indicated that earlier treatment timing had a conspicuously greater suppressive effect than the area treated. Disease spread computer simulations over a broad range of influential epidemic parameter values (R_0_, outbreak disease prevalence, epidemic duration) suggested that potentially unrealistically large increases in treatment area would be required to compensate for even small delays in treatment timing. Although disease surveillance programs are costly, our results suggest that treatments early in an epidemic disease outbreak require smaller areas to be effective, which may ultimately compensate for the upfront costs of proactive disease surveillance programs.

## 1. Introduction

The epidemics caused by long-distance dispersed (LDD) pathogens often begin as one or several spatially distinct cluster(s) of infected individuals (an outbreak focus or foci) from which disease subsequently spreads into the uninfected population and builds in magnitude as the epidemic moves across the landscape. LDD pathogens have a characteristic, highly leptokurtic dispersal kernel (or gradient), which enables the disease to exploit and build rapidly on local hosts (from the steep part of the kernel near the outbreak source) but also to spread across the broader landscape and colonize patchily distributed hosts through the fat, extended tails of the dispersal kernel [1,2]. This type of disease gradient can produce epidemics that increase in velocity at the invasion edge [3,4], generating rapidly spreading epidemics that become difficult to suppress as they gain momentum and overcome dispersal barriers and host spatial heterogeneity [5,6,7,8,9,10]. An accurately described dispersal kernel is essential for developing predictions of epidemic spread. However, the philosophical and analytical approaches for constructing disease spread projections are diverse and range from theoretical to phenomenological [10,11,12,13,14], portraying the lack of consensus amongst epidemiologists. Disease spread models with dispersal kernels based on power law functions have been successfully used to characterize the landscape level spread of human, animal, and plant LDD epidemics [6,9,15,16,17,18]. Not only do power law functions capture the highly leptokurtic disease gradients of LDD pathogens, but they also have a scale invariance property that can simplify forecasting and provide a shared perspective through which human, animal, and plant LDD epidemics may be compared, assuming proportionally scaled conditions and host distribution [6,9,15,16,17].

Dispersal kernels and demographic parameters interact to generate projections of disease spread across the susceptible host landscape. In epidemiology, the basic reproduction number (R_0_), the average number of secondary infections produced from a single infection, reflects the demographic potential for disease increase and is fundamental for understanding epidemic spread [19,20]. For epidemic-causing diseases, R_0_-values exceed 1, the minimum threshold value necessary for disease persistence, often by more than several-fold [21,22,23]. Diseases with larger R_0_-values have greater explosive growth potential and can be difficult to control because any breach in containment confers a greater risk of generating a new, rapidly growing outbreak [19,24,25,26]. When host densities are high and relatively homogeneous relative to the pathogen’s dispersal gradient, as is often the case in dense human and agricultural populations, the combination of long-distance dispersal and high R_0_-values poses considerable challenges for epidemic abatement [25,27,28,29,30].

When a disease outbreak is discovered, area-based treatments (ring vaccination, quarantine, ring culling, etc.) may be administered in hopes of confining the outbreak and preventing disease spread beyond the control area boundary. Typically, such control efforts are limited to the outbreak focus—the region where the disease was first observed—and a variably sized buffer extending into the asymptomatic, potentially latently infected, at-risk population [7,14,31]. This strategy should eradicate the epidemic if the treatment area exceeds the pathogen’s dispersal gradient at the time of establishment. In practice, however, epidemics caused by LDD pathogens are not so easily controlled and often break area-based containment measures [30,31,32,33,34,35,36], indicating that we do not yet sufficiently understand how to design area treatments that consistently suppress outbreaks of LDD pathogens.

Even if epidemiologists could accurately and precisely predict where, when, and how far disease had spread from the outbreak by the time on-the-ground control measures were administered, some area treatments would still remain financially and/or geopolitically impossible to implement despite their efficacy [25,27,30,35]. Because epidemic spread is a spatio-temporal process, the relative success of disease control measures will necessarily be an interaction between the timing of treatments and over how large an area they are implemented. Considerable gains have been made in understanding the spatial patterns of epidemic spread and projecting the effects of area treatments on disease reduction and confinement [27,30,32,34,36,37,38]. These models of epidemic control employ spatially explicit treatment solutions based on the observed amounts of disease and do not explicitly take into account treatment timing. Timing is critical, though. As local disease builds over time, the sheer number and frequency of infectious agents dispersing from the origin grows, substantially increasing the expanse, incidence, and rate of disease spread (including the latently infected hosts) [30,39]. Rapid disease intensification and increased frequency of more long-distance dispersal events over time must influence the area dimensions required to suppress a disease outbreak [30,37]. Nonetheless, the temporal aspects of LDD pathogen control are not particularly well-understood, especially as they relate to the early outbreak, from either empirical or in silico perspectives [30,37,38]. In naturally occurring epidemics, there is considerable spatial and temporal uncertainty associated with the outbreak itself, limiting our grasp of how the outbreak may be related to the later patterns of epidemic spread [15,30]. Understanding how elapsed time since the outbreak beginning influences the size of the effective treatment area may be critical for refining and designing more efficient outbreak suppression strategies. However, such studies will require the tracking of disease spread from outbreaks with known spatial dimensions and unambiguous beginnings [40].

We observed field-initiated outbreaks and disease spread simulations to understand how treatment timing (elapsed time since disease onset) and treatment area (ring cull size) interact to suppress single-focus epidemics of wheat stripe rust (WSR), an important disease of wheat (*Triticum aestivum*) caused by the LDD pathogen *Puccinia striiformis* f. sp. *tritici*. Our emphasis was not solely on the control of WSR per se, but on using this system as a model for studying the impact of epidemic interventions for diseases caused by LDD pathogens. WSR is useful for this purpose, as pathogen lifecycle parameters are well-known; field epidemics can be experimentally initiated in isolation from outside influences and without risk to surrounding agricultural production; disease gradients are well-characterized [41,42]; and the disease gradients are comparable in shape with other LDD pathogens of plants [17,31,43], livestock [7,17,44], and humans [45].

We hypothesized that treatment timing would have a greater overall effect on final epidemic magnitude than ring cull area because earlier treatments will occur at a time before epidemic spread becomes independent of the outbreak [29]. Increases in ring cull radius, however, may compensate for later treatment timings by removing a greater proportion of the latently infected hosts. We sprayed experimentally initiated WSR outbreaks at specific times in the first outbreak generation and with different ring cull sizes to test our treatment timing and area interaction hypotheses under field conditions. Because field experiments were limited to a narrow subset of environmental conditions and range of ring cull sizes, we used simulations to cross-validate the field experiments and evaluate the robustness of treatment interactions over a broader range of environmental and treatment scenarios. To this end, we varied R_0_-values to represent a wide range of known epidemic diseases, modified disease prevalence at the outbreak onset, and increased the number of disease generations. If timing and area treatment effects are robust, the relative treatment influence will be conserved regardless of the simulated epidemic’s growth potential, duration, or outbreak severity.

## 2. Results

### 2.1. Field Experiment

By epidemic end, the comparable portions of the WSR disease gradients suggested that treatment timing exerted a markedly larger relative effect than increased ring cull size (Figure 1 and Appendix A). In particular, the earliest timing (1.125 LP) appeared to substantially suppress disease relative to the later treatments. The most delayed treatment timing, 1.5 LP, had little impact on the final epidemic, regardless of ring cull size, while the 1.25 LP treatment appeared to have an intermediate effect in disease reduction (Figure 1 and Appendix A). There was little evidence that ring cull size influenced the final epidemic severity, even within treatment timings, as we observed no consistent rank order of disease reductions related to increases in ring cull size (1×, 3×, and 5× focus widths) (Figure 1 and Appendix A). The only potential exception to this pattern may reside within the 1.125 LP treatments, where there appeared to be a reduction in disease with greater ring cull sizes (Appendix A). WSR disease was not eradicated by any treatment combination in our field experiments.

When we compared the area under the disease gradient (AUDG) (calculated in all cases for the region of the susceptible hosts outside the largest ring cull), we found strong statistical support in the LMEM for an effect of treatment timing but not ring cull size (Table 1). There was no statistical support for any interaction between timing and ring cull size (Table 1).

### 2.2. Simulations of WSR Spread Approximating Field Conditions

Simulation of WSR disease spread, under conditions approximating those of the field experiment, suggested that earlier treatment timings would have a greater influence on epidemic suppression than increases in ring cull size (Figure 2; Appendix A). Timing treatments at 1.125 LP had the greatest effect on diminishing the final epidemic magnitude, regardless of ring cull size, and the relative amounts of disease increased with later treatment timings. Ring cull treatment effects were consistent within all timing treatments in that increases in ring cull sizes, from 1× to 5× focus widths, slightly diminished the final WSR epidemic magnitude. However, the effect on epidemic suppression through increased ring cull size was a fraction of that imparted by treatment timing (Figure 2; Appendix A).

### 2.3. Robustness of Timing and Ring Cull Size Effects

When we extended the simulations from five latent periods to eight latent periods, we observed an increase in disease abundance and expanse (as expected), but the relative treatment effects remained consistent; i.e., there was a strong timing effect on WSR epidemic suppression and a negligible effect due to increased ring cull size (Figure 2; Appendix A). We observed a similar pattern of WSR suppression for timing and ring cull sizes when we varied basic reproduction numbers (R_0_) from 4 to 140. In these simulations (both at five and eight latent periods), early treatments again had a relatively large effect on final epidemic magnitude, whereas increased ring cull size generated a small effect (Figure 3). When we varied the amount of disease at the outbreak onset (0.5%, 5%, 15%), we observed an increase in final epidemic magnitude with an increase in outbreak disease prevalence (Figure 4). Despite the potential for outbreak disease levels to modify the relative influence of treatment timing and ring cull size effects, treatment timing still exerted a stronger degree of suppression on epidemic magnitude than increased ring cull size (Figure 4; Appendix A).

When we initiated simulations with small amounts of disease (0.0025%) and allowed WSR to build in the focus to the starting levels in our field experiment (1.2%) before applying treatments (45 days, 2.81 LP), each combination of timing and ring cull area minimally impacted the final epidemic magnitude (Figure 5). The rank order of timing treatments, from early (1.125 LP) to late (1.5 LP), and size of ring cull, from 1× to 5×, on WSR suppression was consistent with previous simulations, but the overall effect of any treatment was small.

### 2.4. Interactions between Treatment Timing and Ring Cull Size

The response landscape projected through non-parametric multiplicative regression (NPMR) of WSR disease spread simulations, taking into account disease levels inside and outside of the treatment area (inclusive disease reduction, IDR), suggested a potential compensation for a delay in treatment by an increase in ring cull area. The interpolated response landscape over all initial disease levels (0.05%, 0.5%, 5.0%) consistently projected a large suppressive effect of the earliest treatments (1.125 LP), which became incrementally more effective at reducing disease as ring cull size increased (Figure 6, blue landscapes). There was a plateau of relatively large suppressive effects associated with the larger ring cull sizes (>80 focus widths) and earlier timings (<1.35 LP), with a steep gradient of progressively less effective disease reduction converging on late timing (2.0 LP) and small ring cull size (1× focus width) (Figure 6, blue landscapes). In terms of inclusive disease reduction, our simulations suggest that delayed treatments could be offset by a larger ring cull area. However, this compensation occurred because of host death (removal) within the larger ring cull areas.

The projected response landscape generated by comparing the amount of disease reduced outside of the treatment area (percentage disease reduction, %DR), also suggested that earlier treatment timings would likely be the most effective and efficient at suppressing a WSR epidemic (Figure 6, green landscapes). Unlike the IDR landscapes, there was not strong evidence indicating that a delay in treatment could be offset by increasing the ring cull area. Instead, disease suppression diminished sharply with increasingly delayed treatments, regardless of ring cull area. By 1.4 LP, the IDR response landscape suggested that only small gains in suppressive effect would occur by increasing ring cull area up to 120 focus widths (Figure 6, green landscapes). Projections of delayed treatments with ring culls up to 300 focus widths were nearly identical to the observed effects at 120 focus widths (unpublished data). This response was due to the ring cull size extending into the long, thin part of the disease gradient, where minimal gains occur because disease is low but present at very low levels.

For both disease reduction relativization approaches, an increase in the amount of disease at the outbreak onset resulted in reduced disease suppression by the epidemic end. Overall, the landscape response shapes remained largely consistent within each relativization method over the 100-fold range of outbreak disease levels (Figure 6).

### 2.5. WSR Eradication Potential

Stochastic models of WSR spread suggested that disease would be dispersed far beyond the 5 focal width ring cull treatment at 1.125 latent periods. There was a 100% probability (from 300 stochastic simulations) that multiple effective spores would occur beyond the 5× ring cull treatment, and a 3% probability that a ring cull of 65 focus widths would be able to eradicate WSR with the earliest treatment timing. To have at least an 80% chance of eradicating WSR at 1.125 LP (given a relatively low R_0_ of 4), ring culls would have to exceed 120 focal widths.

## 3. Discussion

Given the reliance on area-based disease control measures, understanding whether and how treatment timing enhances area treatment effectiveness is important for optimizing epidemic outbreak control strategies [7,16,25,31,46]. The WSR disease gradient follows a power law function [41,42], and there are two useful applications of the scale invariance property that are relevant for understanding and projecting epidemic spread. First, the landscape level spread of WSR can be expressed in multiples of focus width because the size of the final epidemic and the focus dimensions scale with each other [15,47]. Second, and perhaps most important for power law distributed pathogens [6,9,15,16,17,18,48], the mechanisms acting at smaller spatial scales (e.g., experimental plots) may also apply to larger spatial scales, assuming host distribution and conditions scale more or less proportionally [6,9,15,49,50]. Thus, the patterns derived from the smaller outbreak field experiments are likely to be relevant for larger, naturally occurring outbreaks. Although naturally occurring epidemics can arise from multiple outbreaks and move through a more complex, heterogeneous, at-risk population [7,12,14,33,34,51], our studies of single-focus disease spread into a continuous host population made evaluation of timing and area treatment effects relatively straightforward and testable in field experiments.

When we evaluated the magnitude of WSR spread outside of ring cull treatments, both the field experiment and simulations were generally consistent in indicating that earlier treatment timings are likely to confer a greater suppressive effect on the final epidemic magnitude than delaying treatments and increasing the ring cull size. Simulations suggested that incremental suppressive effects from increased ring cull size (3× to 5×) were possible, but the field experiment was equivocal in this regard (Figure 1 and Figure 2). Although, the field experiment did suggest a potential effect associated with ring cull size increases when treatments were applied at 1.125 latent periods, between-replicate plot variation was too great to statistically confirm a ring cull size effect (Table 1). Furthermore, we noted that wheat plants in some plots appeared to be more vigorously growing than in other plots, which increases WSR severity. The control plots appeared to have the least vigorous plants of all treatments, and the other treatment plots varied as well, mostly towards more verdant wheat. While we are not fully certain, the differences in wheat vigor between plots (Appendix A) likely generated the wide variation within and between replicate treatments in the field experiment. Simulations and the field experiment were also consistent in suggesting that any combination of ring cull treatments (1×, 3×, 5×), if applied at 1.5 LP or later, would be highly unlikely to reduce the WSR end epidemic magnitude in a meaningful way. Treatments at 1.5 LP did appear to suppress the end epidemic magnitude in simulations, but the 1.5 LP treatments did not statistically differ from untreated outbreaks in the field experiment (compare Figure 1 and Figure 2). This difference may be explained, fully or in combination, by a mismatch between the actual and applied spore production curve, dispersal kernel parameters in the simulations, or environmental/host variation imparted by field conditions within and between plots of the susceptible hosts. Regardless of the differences between simulations and the field experiment, both approaches were consistent in suggesting that later treatments with smaller areas are unlikely to generate a substantial suppressive effect on the WSR epidemic.

Empirical and in silico studies suggest that the outbreak region itself should have a strong influence on the subsequent epidemic magnitude [5,29,30,48,52]. Early treatments may be more effective at suppressing epidemic spread because they can delay disease build-up and reduce the incidence of long-distance dispersal events that accrue with an increase in local disease levels [37,39,48]. Although WSR epidemics were marginally suppressed in two of the 1.125 LP-1× field plots (Appendix A), overall, the 1.125 LP treatments reduced disease levels to a degree that the disease gradient characteristic of an epidemic was not observed (Figure 1; compare 1.125 LP with control gradients). Assuming that the simulations sufficiently approximated experimental field conditions, it is likely that the 1.125 LP treatments suppressed disease because only a relatively small number of the effective spores had been produced and dispersed from the focus by the treatment time. The spore production curve used in our simulations suggested that only ~10% of the total number of estimated spores (for the second latent period) would have been produced by the 1.125 LP treatment timing (Appendix A). With progressively delayed treatments, a greater proportion of spores were likely released from the focus. At 1.25 LP, we estimated about 35% of the estimated spores had been produced, but by 1.5 LP, ~85% had already been produced and presumably dispersed. Ostensibly, the 1.5 LP treatment seems relatively early in the disease cycle. However, by this time the majority of spores were likely released, ultimately generating disease curves that were more similar to those produced by untreated outbreaks (Figure 1 and Figure 2). We did not measure spore production and lesion development in the field, but the qualitative ranking of end epidemic disease levels coincided with what would be expected based on the amount of spores produced at each latent period treatment. The earliest (1.125 LP) treatments had the greatest effect on disease suppression, and the 1.25 LP treatments were intermediate to the earliest and latest (1.5 LP treatments), which did not statistically differ from untreated epidemics. In a manner analogous to invasion potential from introductions/outbreaks with smaller numbers of colonizing propagules [37,39,53], earlier treatment timing is likely to limit the distance and frequency that a disease can be dispersed from the outbreak.

When we considered the disease occurring on hosts inside and outside of the treatment area (IDR), a delay in treatment timing could be countered by large increases in treatment area. The longer the delay in treatment, the larger the treatment area necessary to gain comparable levels of suppression gained through earlier and smaller treatment areas (Figure 6 blue landscapes). In this tradeoff scenario, disease is suppressed by killing (or removing) hosts over an expanse that spans the steep portion of the disease gradient near the outbreak and a significant portion of the disease gradient’s tail. While effective at reducing the total amount of disease across the broader landscape, larger ring cull areas will incur a more substantive economic burden, both for implementation and potential yield losses. Conversely, when we considered only the disease occurring outside of the treatment area (that is, only on living, susceptible hosts), a different relationship between treatment timing and area emerged. Early treatments again were the most effective at suppressing the WSR epidemic, but it did not appear that a delay in treatment could be as readily compensated for by increasing treatment area (Figure 6 green landscapes). This likely occurs because the influence of the outbreak wanes quickly following the outbreak onset, eventually generating an epidemic that spreads independently of the outbreak [29]. Both disease response relativization methods consistently predict that early 1.125 LP treatments would generate the greatest epidemic suppressive effect.

Early identification of epidemic outbreaks is often difficult, in large part due to the low detectability of a relatively small number of infected individuals within a large host population [46]. For example, it has been estimated that initial prevalence of foot-and-mouth disease on a livestock farm is from 5% to 10%, while minimum detection levels range from 10% to 30% [44]. Outwardly, the experimental WSR outbreaks appear similar to WSR outbreak foci observed in commercial wheat fields in late winter/early spring. In both instances, lesions occupy a spatially discrete region, where they appear dense and locally aggregated as they would be in a typically defined focal outbreak [1]. However, visually apparent lesion aggregation does not rule out the possibility that when naturally occurring outbreaks build from small levels of infection over several generations (e.g., to the 1.2% cover in our field experiments), spores and latent infections are dispersed over a much broader area than the detected aggregation. When we ran simulations of timing and area treatments after allowing WSR outbreaks at very low levels of disease (0.0025%) to build to 1.2% (the mean outbreak WSR prevalence from our field experiments), all timing and area combinations were ineffective at suppressing the epidemic (Figure 5). These results are consistent with WSR field experiments where we sprayed the outbreak at 2.5 LP and did not detect any statistically significant differences between the treated and untreated outbreaks [29]. WSR simulations and field experiments suggest that for early, small footprint treatments to effectively suppress a disease outbreak, they must occur shortly after hosts become infectious in the initial disease generation, regardless of outbreak disease levels. A delay in treatment, even when outbreak disease levels are very low, appears to allow spores to effectively disperse beyond that of the perceived focus into the at-risk population, where disease builds and spreads independent of the outbreak itself [29,48].

Epidemic diseases characterized by long distance dispersal are notoriously difficult to eradicate [2,7,9,24,30,31,32,33,34,35,36,53], and we too experienced similar instances of poor disease suppression, even with the early timing treatments and precise knowledge of outbreak locations. In the WSR field experiment, instances of ineffective disease suppression are almost certainly explained by spore dispersal extending beyond the ring cull boundaries at the time treatments were applied—a common phenomenon in failed epidemic control attempts [30,34,37,38,46]. Simulations suggested that we were far from being able to eradicate WSR with any combination of timing and ring cull treatments in the field experiment. Spatially stochastic dispersal simulations projected that ring culls would need to have a radius >65 focal widths and be applied at 1.125 LP to possibly eradicate disease from ~3% of the outbreaks (1.5 m × 1.5 m) at 1.2% initial disease prevalence. The consequences of delayed treatment were quite clear, as a containment buffer radius of >670 focal widths would be required if the treatment timing occurred at 1.5 LP. Similar to other disease epidemics, the footprint of a treatment with a high probability of WSR eradication would be excessively large and may not be financially, economically, geopolitically, or logistically possible to implement as the time since the outbreak increases [25,27,28,30,46].

Our findings suggest that early treatments, even with a relatively small footprint, can generate a relatively large and substantial suppressive effect on epidemic spread, reinforcing the calls from epidemiologists to move towards more intensive surveillance programs that increase detection probability, facilitate rapid mobilization, and define target areas that maximize epidemic suppression [24,25,46,54,55]. It is quite possible that the suppressive responses to treatment timing and area we present for WSR are similar for other LDD diseases. Simulations suggest that the relative effects of treatment timings and their interactions with ring cull size may be conserved over a broad range of outbreak disease levels (0.5% to 15%), R_0_-values (4 or 140), and increases in epidemic duration—all of which span the range of values documented for human, epizootic, and plant LDD diseases. Although our study focused on the suppression of epidemic spread from a single outbreak, these responses to timing and area treatments may be relevant for scenarios of multiple outbreaks and sequential treatment of subfoci that escape initial treatment.

## 4. Materials and Methods

### 4.1. WSR Study System

*Puccinia striiformis* f. sp. *tritici*, the causal agent of WSR, is an obligate parasitic fungus of wheat (*Triticum aestivum*) that requires high night-time relative humidity, mild temperatures, and susceptible host plants for infection, lesion growth, and sporulation [56]. Unlike necrotrophic pathogens, which capitalize on weakened hosts, WSR virulence, spore production, and lesion growth all diminish with reductions in host vigor [56]. Hundreds to thousands of passively dispersed urediniospores (oblong spheres ~25 μm × 20 μm) are produced daily over the infectious period from individual pustules [57]. Aggregate pustules appear en masse as rust-orange lesions, borne as linear arrangements on the upper and lower leaf surfaces, often covering a large percentage of the leaf surface. WSR’s latent period—the time from infection to sporulation—varies from 10 days to several weeks, depending on host physiological condition and abiotic conditions such as temperature, humidity, and water availability. In most instances, naturally occurring WSR epidemics persist for 4–5 latent periods, but they occasionally span 7–8 latent periods in warmer, more humid, wheat-growing regions of the world [52,58,59].

We initiated single-focus, WSR outbreaks in wheat fields without risk to agricultural production through the planting of heirloom, non-production cultivars and the use of WSR pathogen races to which all contemporary production cultivars have complete resistance [29]. Field experiments took place in a small, geographically isolated arid growing region of central Oregon where the low winter temperatures (mean high temp December = 5.5 C, mean low temp December = −3.4 C) are unfavorable for WSR overwintering, together limiting the incidence of non-experimental outbreaks.

### 4.2. Ring Cull and Timing Treatments

Preliminary epidemic simulations (unpublished) allowed us to estimate a range of timing treatments that would be both biologically relevant for disease control and practical to implement. These exploratory simulations suggested that we would be unlikely to detect an impact of culling after 1.5 latent periods (a half generation of spore dispersal following the first day of outbreak sporulation). We confirmed this hypothesis with two field experiments: one demonstrating no difference in the final epidemic magnitude between treated and untreated outbreak foci (ring cull size = focus size) at 2.3 latent periods after inoculation with WSR spores (LP) [29] and another similar experiment with no differences in final epidemic magnitude between treated and untreated foci at 1.5 LP (unpublished data). From these preliminary simulation and field studies, we chose three timing treatments that were at and prior to 1.5 LP for our experiment (1.125 LP, 1.25 LP, 1.5 LP).

For ring cull sizes, preliminary simulations failed to identify any size of realistically implementable ring cull treatment (given our experimental plot dimensions) that would likely reduce the final epidemic magnitude by >30% when compared with a treatment of just the outbreak focus. Thus, we chose ring cull areas of 1× (focus only), 3× (focus + 1 focus-width buffer), and 5× (focus + 2 focus-width buffer), which equaled the experimental plot width. The 1× and 3× culls are analogous to the infected premises (IP) and contiguous premises (CP) ring culling used for the control of foot-and-mouth disease of UK livestock [7], respectively; the 5× area treatment tests for the advantage, if any, of increasing the buffer size beyond a contiguous premises cull. We express the dimensions of area treatments in focus widths because the final magnitude of experimental WSR epidemic scales to the outbreak focus dimensions [47]. This method of relativizing invokes the scale invariance property of the power law to extrapolate potential effects to larger spatial scales [6].

### 4.3. Field Experiment with Factorial Timing and Ring Cull Combinations

Our field experiment was conducted on private property near Culver, OR, USA. Because this agricultural region is arid, weekly irrigation of experimental fields was necessary to encourage vigorous host growth and epidemic levels of WSR. Fields were planted with two cultivars: one susceptible and one resistant to the experimental WSR race PST 29 [56]. Cultivar Jacmar is highly susceptible to PST 29 and was planted in 7.62 m × 41 m plots, with the long axis oriented east–west (parallel to west-northwest prevailing winds) [60]. Surrounding these experimental plots were buffer areas, >30 m between experimental plots on all sides, that were planted with the cultivar Stephens (completely resistant to pathogen race PST 29). Inoculation of outbreak foci (1.52 m × 1.52 m) occurred 7 m from the western end of the experimental plots to generate a downwind plume of disease.

PST 29 spores were collected from infected wheat plants grown in indoor growth chambers [29,41]. We used a spore–talc mixture of 0.25 g spores and 3.75 g of talc to inoculate each focus [29,41]. Inoculation took place on the evening of 17 April 2014, with calm winds and overnight temperatures that were mostly above freezing, except the hour before and after dawn. The spore–talc mixture was applied to pre-moistened plants inside a PVC pipe frame covered with clear plastic film to prevent spore escape. To maintain high overnight relative humidity, which increases infection efficiency, confines spores to the focus, and shelters plants from potential overnight frost, we immediately covered the inoculated area with black plastic sheeting for the following 13 h [41].

We inoculated foci in 40 total plots and assigned a full complement of treatments (described below) to four blocks within large regions of the field arranged by both north–south and east–west orientations. We used a plot arrangement design within each of four field regions (blocks which contained a replicate of all treatments), such that we could evaluate potential field region effects that were due to an unforeseen environmental gradient if viewed from a north–south perspective or an east–west perspective. This nested random arrangement of experimental plots was used to evaluate the potential interactions of environmental conditions (e.g., soil richness, irrigation patterns, slope, fertilizer, drought stress, etc.) within specific regions of the field. The capacity to statistically evaluate region effects due to east–west and north–south arrangements was the most thorough way to account for larger scale spatial differences that we could not have foreseen during the fall planting and spring growing seasons. Spray treatments were a mixture of fungicide and herbicide: Stratego^TM^ (trifloxystrobin + propiconazole), which stops fungal lesion growth and spore production within hours; and glyphosate, which initially reduces plant vigor and eventually leads to host death or reduces vigor to a degree which WSR can no longer survive. The nine treatments consisted of factorial combinations of three treatment timings (1.125, 1.25, and 1.5 latent periods) and three ring cull areas (1×, 3×, 5× focal widths), plus an inoculated, unsprayed control. Each treatment was replicated four times, once in each block. We used local weather station data, observed timing of first lesion expression, and a refined WSR growth degree-day model [61] to determine latent period spray timings. Ring cull sprays were square, like the inoculated focus, with width and length of 1.52 m (1×), 4.57 m (3×), and 7.62 m (5×).

We recorded disease prevalence, as a percentage of the maximum possible number of lesions (~100 lesions/tiller), in 1.52 m × 1.52 m quadrats centered over the focus and at 1.5, 3, 6, 9, 12, 15, 18, 21, 24, 27, and 30 m eastward (downwind) from the outbreak focus. We began disease assessments at ~3.8 latent periods and continued collecting data weekly until the epidemic ended in early July, at ~5 latent periods. Disease ratings were delayed until the later generations to limit any potential inadvertent disease dispersal by observers navigating within plots to record disease in the earlier generations. In addition, the delay enabled us to quantify potential treatment effects at maximum disease severity, which in WSR on wheat fields in our study location occurs just prior to grain filling. Observer bias in disease prevalence estimates was reduced through cross-calibration and recording the mean disease prevalence value from the same observer pairs.

For each field, we plotted disease prevalence (*y*-axis) against distance from the focus (*x*-axis), creating a disease gradient. As an estimate of final epidemic magnitude, we calculated the area under the disease gradient (AUDG) on the last assessment date with the mid-point method [62]. To compare treatments with different ring cull sizes, we used only the AUDG-values beginning with the first subsampling location outside of the largest cull area (~6 m from the focus center). To determine whether final epidemic magnitude differed among treatments, ln-transformed AUDG-values were analyzed with a linear mixed-effects model (LMEM), with spray timing and treatment area as fixed effects and block as a random effect using Proc GLM in the program SAS [63]. Due to inconsistency in host plant health within some plots, which rendered them inappropriate for statistical comparison, we omitted one of each of the following replicates from the final model: 1.125 LP (latent periods)- × (focus widths), 1.25 LP-5×, 1.5 LP-3×, and 2 replicates for 1.5 LP-5×. Initial analysis indicated that there was no statistically significant treatment effect of the block region in the field (*p* > 0.50) for both east–west and north–south block arrangements on AUDG-values due to region, so we did not include “region” as an effect in the final model.

### 4.4. Cross-Validation of Field Experiments with WSR Spread Simulations

We used a highly modified version of the program EPIMUL [64,65], a SEIR epidemiological model, to cross-validate the field experiments by simulating WSR disease spread under virtual conditions that approximated those of the field experiment (e.g., latent and infectious periods, R_0_, spray timings, disease outbreak levels, focus size, carrying capacity, disease gradient). EPIMUL distributes effective spores over the simulated area using an inverse power law function over a virtual landscape comprised of square host compartments, which can be populated with variables related to WSR reproduction, growth, and infectious and latent periods, as well as characteristics of the host population (carrying capacity, resistance, density, etc.) (see [65] for a detailed description). Within each compartment, the probability of effective spores producing a lesion is proportionally decreased as disease builds and the number of available infection sites is lowered.

We selected a base set of parameters to represent the field experiment conditions (most notably R_0_, outbreak disease levels, and plot dimensions), then modified some variables to estimate treatment effects over a range of conditions beyond those of the experiment. Our default virtual field was 1000 × 1000 compartments, each 1.52 m × 1.52 m in size—the same size as the outbreak focus in the field experiment. Compartments of this size enabled us to evaluate the effects of treatments in terms of focus multiples due to the scaling relationship between WSR outbreak focus dimensions and final epidemic magnitude [47]. Although the virtual field was much larger than the experimental plots, a larger virtual field was necessary for exploring treatment effects for larger ring culls and longer duration epidemics. Each compartment had a carrying capacity of 200,000 lesions (based on average WSR field values and host plant densities from multiple studies) and effective spores were distributed along the power law function y = *a* ∗ (*x* + *c*)^−*b*; where y = the number of effective spores, *a* is a constant that is proportional to the amount of source inoculum, *x* is the distance from the source lesion, *b* is the steepness of the curve, and *c* is an offset parameter than enables a finite value at *x* = 0 [65]. We used the dispersal parameter values *b* = 2.28, *c* = 0.23, and *a* = 428 for all simulations, as these values represented the most accurate and precise dispersal gradient available to us (derived from intensive studies of disease dispersal from group and single lesions [41,42]). We placed emphasis on the dispersal gradient because seemingly subtle changes in the gradient’s shape and width, as well as truncation of the distribution’s tails, is well-known to generate erroneous and inaccurate dispersal projections [2,3,4,56,65,66].

Unless stated otherwise, we used the average outbreak disease prevalence (1.2%, or 2400 of the 200,000 infection sites in EPIMUL) of the four unsprayed, control treatments from the field experiment as the outbreak disease level (P_0_) in disease spread simulations. We estimated R_0_ by holding known epidemic variables constant (P_0_, dispersal gradient, and latent period) and finding the value of R_0_ that produced a final simulated epidemic closely matching the epidemics in the unsprayed field plots at 5 latent periods. This yielded a R_0_ estimate of 70, similar to an independently derived estimate of 69 reported by Segarra et al. [67] for WSR in Europe. We used a latent period and infectious period of 16 days based on climate conditions, degree-day models, and field observations gathered over the duration of the field study. Although the WSR latent period may vary with temperature [61] and the field conditions did fluctuate with some temperatures being warmer or colder depending on the week, the +/−1-day weekly variation from the overall 16-day latent period caused by weather in the field experiment is unlikely to substantially influence our simulation results in a manner that would qualitatively alter our interpretation of the relative influence of timing and area treatment effects [65].

EPIMUL was reprogrammed to halt effective spore production in specified compartments, analogous to ring cull areas on a specified day [32]. EPIMUL produces spores at the same rate on every day of the infectious period. WSR spore production has been shown to vary over time [68], so we used the WSR spore production curve presented in Papastamati and van den Bosch [68], which most closely matched our field experiment temperatures, to estimate the amount disease in the focus at each spray timing (Appendix A). In applying this curve, we assumed that spore production, lesion production, and the number of effective spores are sufficiently positively correlated to consider the spore production curve a suitable proxy for the proportion of lesions treated at the three different spray latent period timings (1.125 LP, 1.25 LP, 1.5 LP). Except for specific scenarios (described below), simulations were run for 5 latent periods (to be consistent with the field experiment duration) and other variables such as lesion growth, spore infection efficiency, host genetic resistance, number of infection sites/compartment, and number of pathogen races were unvaried among all simulations.

It is possible that ring cull treatments could be overwhelmed by environmentally driven stochastic variation. To address this issue, we created a version of EPIMUL with spatially stochastic spore dispersal as a means to simultaneously approximate the environmental and host variation that ultimately generates differences in the spatial distribution of disease over a continuous distribution of susceptible hosts. Effective spores, originally distributed along the power function (y = *a* ∗ (*x* + *c*)^−*b*), were selected from the same disease gradient, but their location along that distribution was randomly redrawn from a Poisson distribution. Doing so conserved both the large amount of near source effective spore dispersal but also generated the longer distance dispersal events we know to occur [39]. Overall, stochastic simulations performed as expected, with occasional increases/decreases in local disease abundance and the infrequent long-distance dispersal of disease beyond the limits projected through deterministic simulations. The variation imparted by dispersal stochasticity was less than the treatment effects in all but 1 of 100 simulations (Appendix A). Consequently, we report the results from deterministic simulations in all but one instance, when it was necessary to assess the possibility of WSR eradication.

### 4.5. Robustness of Timing and Ring Cull Effects

It was necessary to evaluate whether treatment effects were constrained by experimental field conditions or whether they were part of a more inclusive and general response. We used simulations to explore the consequences of changing three variables known to exert a strong influence on epidemic magnitude. First, we varied the basic reproduction number, R_0_, but held all dispersal parameters and the initial disease level constant. By varying R_0_ we assessed the potential range of responses to timing and area treatments while varying one of the more sensitive and unifying variables for describing and predicting epidemic severity [19,21,24]. R_0_ values covered the range of important human (R_0_ = 4 influenza, R_0_ = 20 measles) [6,19], epizootic (foot-and-mouth disease of livestock, R_0_ ~ 5 to 50) [44], and plant epidemic diseases (R_0_ = 70 WSR from our field experiment, and R_0_ = 140 a hypothetical and abnormally severe WSR epidemic).

Second, we simulated WSR spread for an additional three latent periods, allowing disease to accumulate and disperse beyond what we observed in the field experiments. Extending the simulations to eight latent periods represents an abnormally long, naturally occurring, WSR epidemic [52,58,59], over which additional disease build-up might diminish the relative impacts of area and timing treatments.

Third, outbreak disease prevalence (P_0_) can strongly influence epidemic magnitude [39] and thus potentially modify ring cull and timing treatment effects. We ran additional simulations at 0.5%, 5.0%, and 15.0% outbreak disease prevalence, with each factorial combination of ring cull and timing treatments, to determine whether treatment effects may be robust to changes in P_0_.

Last, we considered the possibility that treatment interactions may differ when initial outbreak disease levels (P_0_) are very low but treatments are delayed until disease builds to a more conspicuous level in the focus. To evaluate this scenario, we initiated WSR simulations with a very small amount of disease (0.0025%), allowed the disease to increase until it reached ~1.2% in the focus, and then applied the experimental timing and area treatments beginning on that date (day 45 = 2.81 latent periods, +1.125 LP, +1.25 LP, +1.5 LP). We extended the duration of these epidemics to 7 latent periods.

### 4.6. Interactions between Treatment Timing and Ring Cull Size

To understand the potential interactions between treatment delays and increases in ring cull size, we set up simulations that delayed the spray treatments (1.125 to 2.0 latent periods), increased the treatment area (from the focus only up to a diameter of 120 focus widths), and evaluated the final epidemic magnitude over a 100-fold range of (P_0_) outbreak disease levels (0.05%, 0.5%, 5%). We calculated and compared the final epidemic magnitude resulting from the different treatment variable combinations with two different methods. First, we estimated the percentage of disease reduction (%DR), relative to an uncontrolled epidemic, occurring outside of the treatment area (treatment disease/disease occurring over the same distance in an uncontrolled epidemic * 100). This is the same approach we used to compare epidemics in the field experiment. With this method, we evaluate only the relative effect of the treatment on the host population with the possibility of being diseased (recall our spray treatments killed host plants; e.g., removed in SIR or SEIR models). The %DR-values should reflect any efficient and inefficient tradeoffs between timing and area treatments. Second, we calculated the inclusive amount of disease reduction (IDR), which accounted for disease in both the treatment area and that outside of the treatment area, divided by an uncontrolled epidemic. IDR is an estimate of disease reduction due to both host death (plants in the treatment area) and disease reduction in hosts outside of the treatment area. With this method of relativization, we expected tradeoffs between spray area and timing because larger, later sprays are likely to have similar effects to smaller, earlier sprays, as larger treatments will reduce disease directly from host death. Treatment and area timings where %DR and IDR converge may indicate the most effective and efficient treatment combinations for epidemic suppression.

Because WSR disease spread simulations are time-consuming (~1 deterministic run in a 24 h period) and the time required to run the number of simulations necessary to fully populate a comprehensive matrix of latent periods and ring cull size combinations was prohibitive, we used a smaller number of simulations and interpolated the response landscape between treatment combinations with non-parametric multiplicative regression (NPMR) [69,70] in the program Hyper Niche 2 [71]. We expected to observe steep, local responses between some timing and area combinations but relatively flat responses between others, so we required a dynamic interpolation method that could account for potential non-linear interactions and remain faithful to local variable response topologies. NPMR uses local regression models between points clustered in response space and applies kernel smoothing to these smaller local regressions to construct a larger response landscape based on predictor variable tolerance values, neighborhood size, percentage of the variance explained by the model (xR^2^ in NPMR analyses), and local model regression form (Gaussian in our interpolation model). With NPMR, variables are free to interact non-linearly, interactions between predictor variables do not require prior specification (they are optimized during modeling), and overfitting is accounted for during model fitting [70]. Compared with other landscape response modelling techniques such as LOWESS, NPMR is computationally intensive but does tend to produce more faithful and accurate response landscapes [70].

### 4.7. WSR Eradication Potential

It is possible that the combination of timing and ring cull treatments in the field experiment may be able to eradicate WSR. To predict whether eradication may be possible, we simulated disease spread following ring culls with the three treatment timings (1.125 LP, 1.25 LP, and 1.5 LP), sizes up to 5× the outbreak focus width, in a virtual field considerably larger than the plots in our field experiments. Exploratory stochastic simulations of WSR spread to 1.25 and 1.5 latent periods indicated that disease spread far exceeded that of the largest experimental ring cull size of 5× focus widths (e.g., >670 focus widths, ~1 km, at 1.5 latent periods). Consequently, we evaluated the potential for eradication at only the earliest timing (1.125 LP) and with an R_0_ value of 4 (an extreme WSR scenario), as it was obvious that an epidemic with a greater R_0_ would not be contained by our treatments. A R_0_ value of 4, could, for example, represent an uncommon, hypothetical biological situation where a mostly WSR resistant, but not entirely resistant, wheat cultivar is grown. We ran 300 stochastic simulations to evaluate the possibility that the largest ring cull size (5×) at the earliest treatment timing (1.125 LP) could eradicate a single WSR outbreak of 1.2%.

## Figures and Tables

**Figure 1 pathogens-11-00393-f001:**
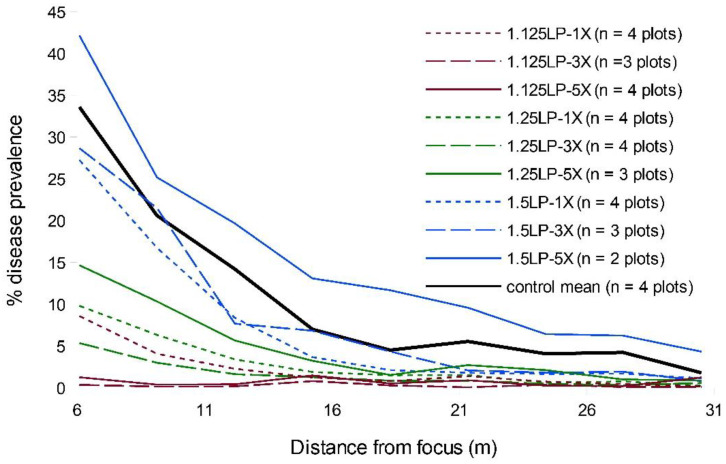
Mean disease gradients of experimentally initiated WSR field epidemics with all pairwise combinations of 1.125 (red), 1.25 (green), 1.5 (blue) latent periods (LP) and 1× (dotted), 3× (hashed), and 5× (solid) focus width ring cull sizes, and untreated control epidemics.

**Figure 2 pathogens-11-00393-f002:**
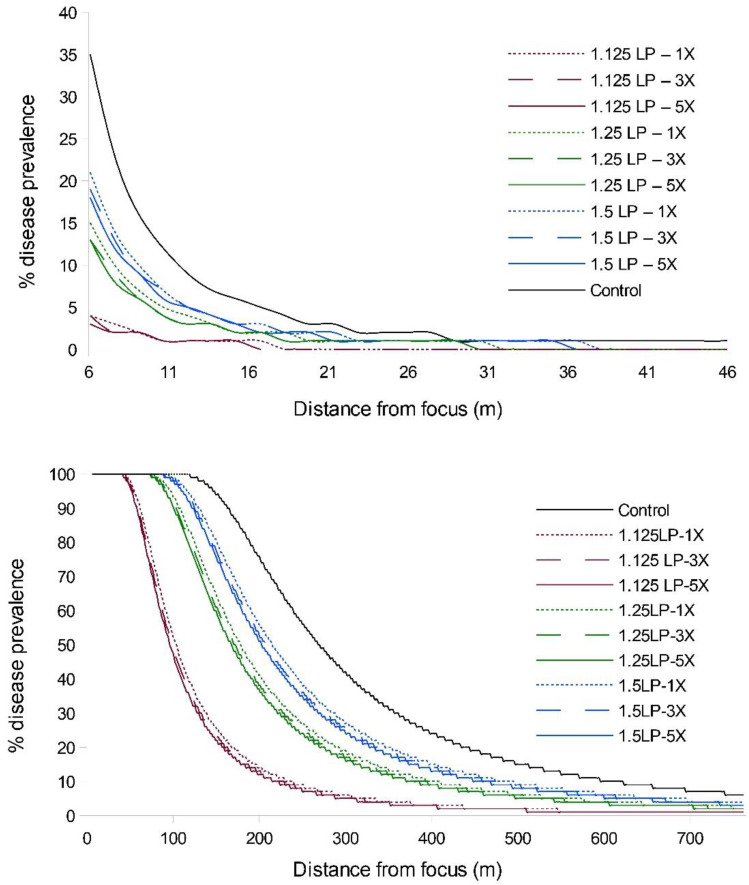
WSR disease gradients from EPIMUL simulations run through 5 (**top**) and 8 (**bottom**) latent periods with parameters approximating field conditions (1.2% initial disease, R0 = 70). Red lines = 1.125 LP, green lines = 1.25 LP, blue lines = 1.5 LP timing treatments; dotted lines = 1× ring cull, hashed lines = 3× ring cull, solid colored lines = 5× focus width ring cull treatments; solid black line = untreated control.

**Figure 3 pathogens-11-00393-f003:**
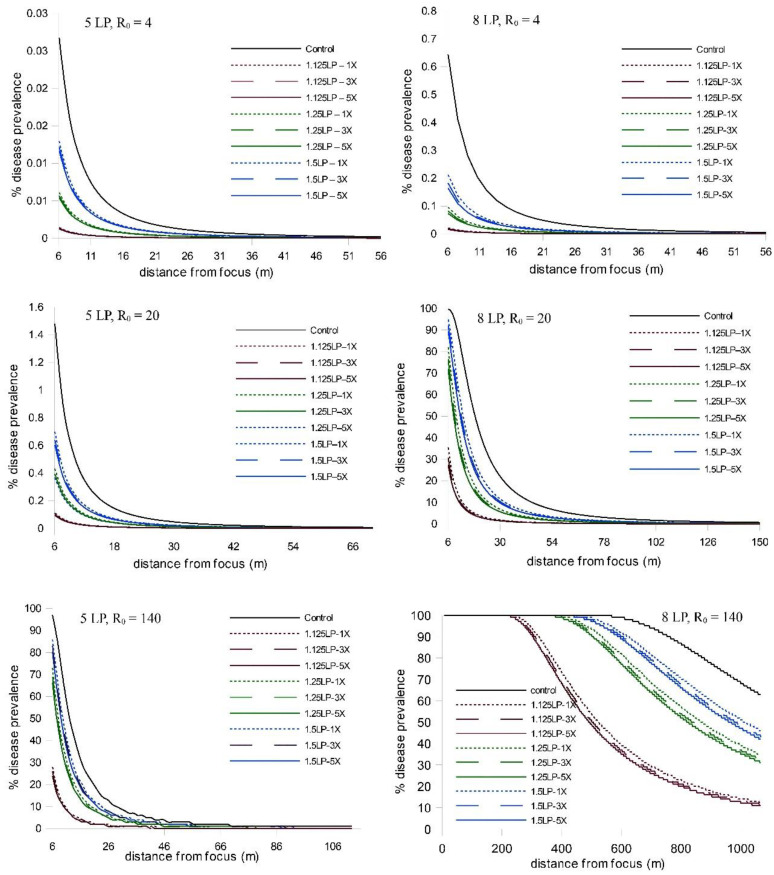
Disease gradients produced from WSR spread simulations after 5 and 8 latent periods (LP) with three different basic reproductive number values (R_0_ = 4, 20, 140). Outbreak disease prevalence (1.2%) and the dispersal gradient were held constant in these simulations. Red lines = 1.125 LP, green lines = 1.25 LP, blue lines = 1.5 LP timing treatments; dotted lines = 1× ring cull, hashed lines = 3× ring cull, solid colored lines = 5× focus width ring cull treatments; solid black line = untreated control.

**Figure 4 pathogens-11-00393-f004:**
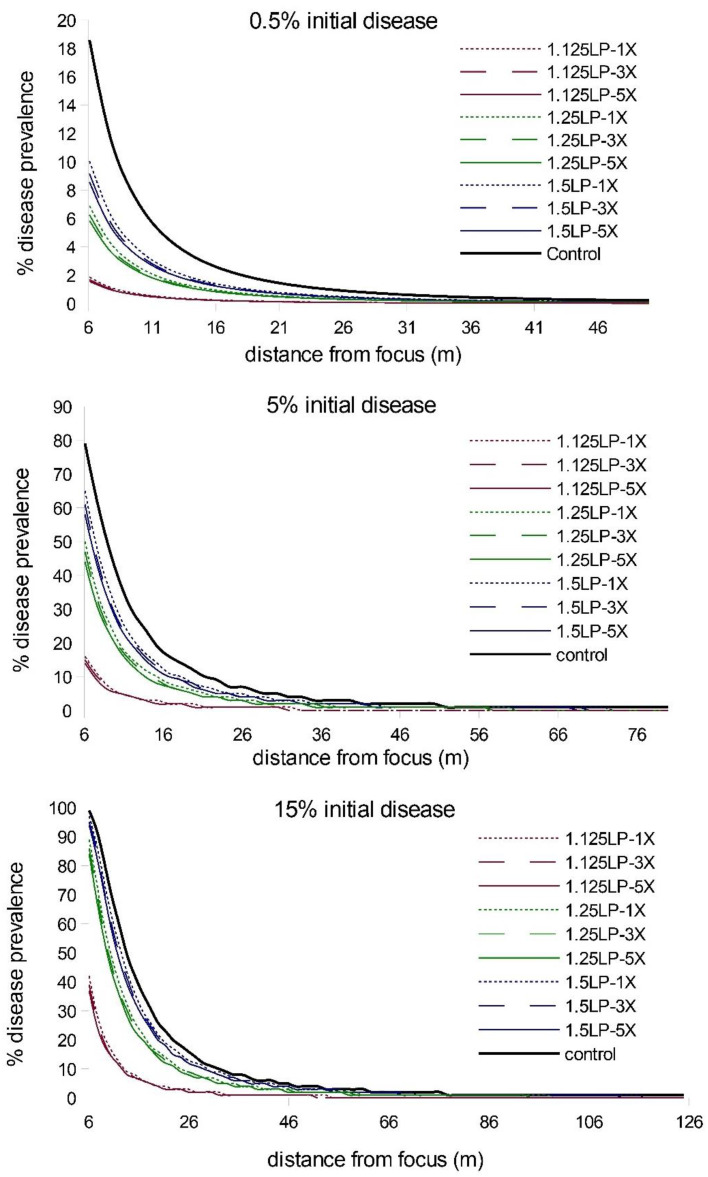
Disease gradients produced from WSR spread simulations after 5 latent periods (LP) for three different outbreak disease levels (0.5%, 5%, 15%) with the basic reproductive number held at R_0_ = 70. Red lines = 1.125 LP, green lines = 1.25 LP, blue lines = 1.5 LP timing treatments; dotted lines = 1× ring cull, hashed lines = 3× ring cull, solid colored lines = 5× focus width ring cull treatments; solid black line = untreated control.

**Figure 5 pathogens-11-00393-f005:**
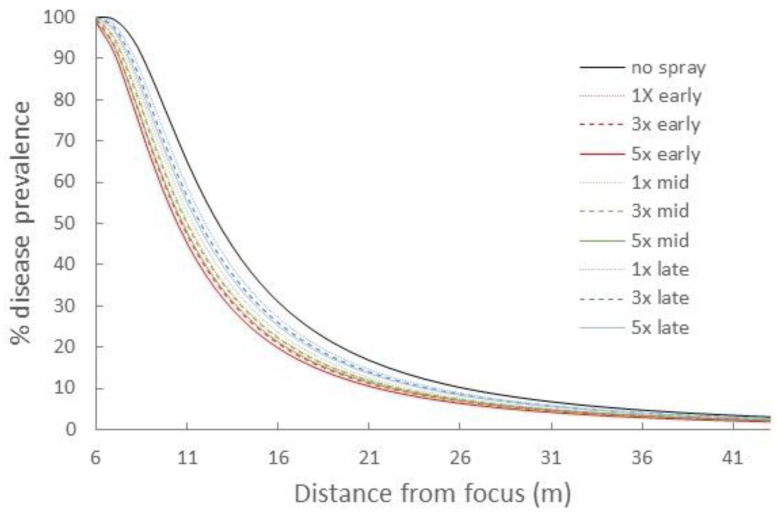
Disease gradients produced from WSR spread simulations after 0.0025% initial disease levels were allowed to build to ~1.2% (day 45, 2.81 LP) in the focus, and treatments were applied beginning from day 45 onward. Red lines = +1.125 LP (early), green lines = +1.25 LP (mid), blue lines = +1.5 LP (late) timing treatments; dotted lines = 1× ring cull, hashed lines = 3× ring cull, solid colored lines = 5× focus width ring cull treatments; solid black line = untreated control.

**Figure 6 pathogens-11-00393-f006:**
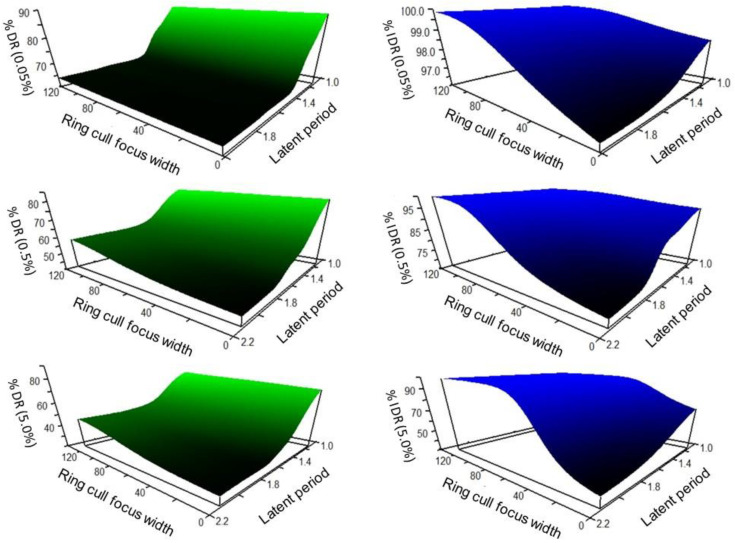
Non-parametric multiplicative regression projected disease reduction response landscapes interpolated from WSR disease spread simulations with disease outbreak levels of 0.05%, 0.5%, 5.0%, treatment timings of 1.125 LP to 2.0 LP, and ring cull sizes of 1 focus width to 120 focus widths. Green response landscapes represent disease reduction in the host population outside of ring culls and the outbreak (percentage disease reduction, %DR). Blue response landscapes represent the disease reduction in the host population including the ring cull and outbreak (inclusive disease reduction, IDR).

**Table 1 pathogens-11-00393-t001:** Linear mixed effects model results for final epidemic AUDG-values for different timing and area treatments.

Effect	*d.f.* Treat	*d.f*. Group	*F*-Stat	*p*
Ring Cull Size	2	24	0.18	0.84
Treatment Timing	2	24	12.99	0.0002 *
Size × Timing	4	24	0.84	0.51

* = *p* < 0.05.

## Data Availability

Not Applicable.

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
