# Peer review of "Delays in Epidemic Outbreak Control Cost Disproportionately Large Treatment Footprints to Offset"

_pathogens, 2022, doi:10.3390/pathogens11040393_

Round 1

Reviewer 1 Report

This is a very well written paper. I like the combination of both experimental and simulation tests. Honestly I can't properly assess the novelty of this study as my expertise is more linked to invasion ecology. However, I feel the results can be of interest to readers, particularly considering the importance of the wheat disease treated in this study. My comments below are mostly related to the methods and results, where I found significant flows in the details and some findings in the experiment which do not make sense. 

L136 – temperature is also an abiotic condition. Please, be more specific

L147 – please, elaborate in this epidemic simulations. Could you provide results in the appendix?

L193: this isn’t very clear. What do you mean with “fields blocked”. From the previous paragraph it seems all plots are surrounded by >30 m of non-susceptible cultivar

L207: why collecting data so late? Any effect of sampling in the dispersal of spores??

L210: There are many aspects not mentioned for the GLM:

  • which error distribution did you use in the GLM? If you apply a Ln transformation it seems you applied a log-normal linear model instead.
  • Also about the region, please clarify this aspect in the previous paragraphs as it isn’t very clear how many plots were placed in each category of “blocking”. 
  • Did you check for assumptions of normality and homocedasticity? 

L249: which value did you use for parameter “a”

L266: to validate with experiment, why didn’t you use the same latent period estimated for the field experiment?

L355: clarify how many simulations you finally carried out.

L380: is this simulation using R0=4 realistic considering that plant diseases usually have R0 one order of magnitude higher (R0=70) than the parameter considered here?

L386-400 Fig1. : the results of cull size treatment seem very strange as it seems that wider sizes (5X) produced higher disease prevalence than lower sizes for each latent period treatment. This is not expected (see simulations in Fig.2), particularly as the treatment 5X and 1.5 latent period showed even higher disease than the control! This strange pattern is not mentioned in the results and it is very likely that there is a problem in the experimental setup or data handling. For example, could it be possible that during the spraying treatment the operators facilitated spore dispersal? Looking again at the results in the stats, it is indicated that there is no significant effect of cull size (Table 1), but probably doing a post-hoc test you will find significant differences between the 1x and 5x treatment.

Author Response

L136 – temperature is also an abiotic condition. Please, be more specific

We have rewritten this passage, it now reads “WSR’s latent period—the time from infection to sporulation—varies from 10 days to several weeks, depending on host physiological condition and abiotic conditions such as temperature, humidity and water availability.”

L147 – please, elaborate in this epidemic simulations. Could you provide results in the appendix?

These simulations were stripped down versions of the ones we conducted and are not suitable to present as they were coarse and rather crude compared to what we produced in the manuscript. Our inclusion of them in the manuscript was to provide a narrative of our steps towards the conceptualization of the study, which often is left out of many studies.  In this case, we thought it important for the reader to have access to our initial rationale for the study.

L193: this isn’t very clear. What do you mean with “fields blocked”. From the previous paragraph it seems all plots are surrounded by >30 m of non-susceptible cultivar

L207: why collecting data so late? Any effect of sampling in the dispersal of spores??

Thank you for bringing these topics up.  We had a specific philosophy in visiting the experimental plots later in the epidemic, namely to evaluate the potential treatment effects when disease is at is maximum expression and to decrease the chances of inadvertent spread due to visiting the plots.  The passage in question now reads. “Disease ratings were delayed until the later generations to limit any potential inadvertent disease dispersal by observers navigating within plots to record disease in the earlier generations. In addition, the delay enabled us to quantify potential treatment effects at maximum disease severity, which in WSR on wheat fields in our study location occurs just prior to grain filling.”  

L210: There are many aspects not mentioned for the GLM:

  • which error distribution did you use in the GLM? If you apply a Ln transformation it seems you applied a log-normal linear model instead.
  • Also about the region, please clarify this aspect in the previous paragraphs as it isn’t very clear how many plots were placed in each category of “blocking”. 
  • Did you check for assumptions of normality and homocedasticity? 

Thank you for bringing to our attention the issue with the analysis of the field data.  What we used was a linear mixed effects model (LMEM), which we have indicated in the revision and omitted reference to a GLM and replaced the description and reporting of the test results as a LMEM in the revised manuscript. We also revised the passage with reference to blocking and regions, and believe the modifications in the revised version are now clear with respect to design.

With respect to the comment concerning tests for normality/skew.  With the low number of treatment replicates we have in this study it is well-known that tests for normality and skew are inaccurate enough to be highly unreliable, so we did not apply them to this specific analysis. We deemed the ln-transformation the best approach to deal with the observed between and within treatment variation in AUDG values prior to running the LMEM.  Unfortunately, these field studies are incredibly difficult, time consuming, and expensive to execute, which is why such studies are relatively uncommon in the literature and unfortunately the numbers of replicates is necessarily low from a statistical perspective.  This is just an unfortunate tradeoff of attempting to replicate experimental disease outbreaks.  We analyzed the data in the most conservative, direct, and responsible way that we could given the limits to the field experiment.

L249: which value did you use for parameter “a”

We indicated that the a-value was 428 in the revision.

L266: to validate with experiment, why didn’t you use the same latent period estimated for the field experiment?

We explained our reasoning in the original version with the following two sentences. “We used a latent period and infectious period of 16 days based on climate conditions, degree-day models, and field observations. Although the WSR latent period may vary with temperature [53], variations of latent period caused by weather are unlikely to influence our simulation results in a manner that would qualitatively alter our interpretation of treatment effects [57].”

In the revision we added a statement that the latent period may differ by one or two days under the conditions of our study.  The manuscript now reads, “We used a latent period and infectious period of 16 days based on climate conditions, degree-day models, and field observations. Although the WSR latent period may vary with temperature by one or two days [53], variations of latent period caused by weather are unlikely to influence our simulation results in a manner that would qualitatively alter our interpretation of treatment effects [57].” Hopefully this clarifies our rationale.

L355: clarify how many simulations you finally carried out.

We indicated in the revised version that there were 84 simulations run for this analysis.

L380: is this simulation using R0=4 realistic considering that plant diseases usually have R0 one order of magnitude higher (R0=70) than the parameter considered here?

No, our point was to consider what treatment effects might we expect over a broad range of epidemiological conditions outside what is known for wheat stripe rust.  It is an evaluation of potential generality of time versus area assuming a similar dispersal kernel and relative host abundance (at-risk population conditions). We report our philosophy in subsection 2.5 Robustness of timing and ring cull effects in the original submitted version.

L386-400 Fig1. : the results of cull size treatment seem very strange as it seems that wider sizes (5X) produced higher disease prevalence than lower sizes for each latent period treatment. This is not expected (see simulations in Fig.2), particularly as the treatment 5X and 1.5 latent period showed even higher disease than the control! This strange pattern is not mentioned in the results and it is very likely that there is a problem in the experimental setup or data handling. For example, could it be possible that during the spraying treatment the operators facilitated spore dispersal? Looking again at the results in the stats, it is indicated that there is no significant effect of cull size (Table 1), but probably doing a post-hoc test you will find significant differences between the 1x and 5x treatment

Thank you for bringing this to our attention.  We were actually acutely aware of the potential for observers to disperse spores through the field which is why we delayed our entry and disease ratings until the last generation of disease. In our revision we make this approach clear as it was not revealed to reader our motivations for taking disease ratings late in the season.  We added the following passage to the methods section in the revised manuscript: “Disease ratings were delayed until the later generations to limit any potential inadvertent disease dispersal by observers navigating within plots to record disease in the earlier generations. In addition, the delay enabled us to quantify potential treatment effects at maximum disease severity, which in WSR on wheat fields in our study location occurs just prior to grain filling.”

In the original manuscript we did not discuss the variation in host plant health within our experimental plots.  For wheat stripe rust, disease is much more severe on plants with increasing vigor.  One of the aspects that was out of our control for this experiment, was the observation in the spring (wheat is planted in the fall of the previous year) is that some plots produced much more vigorous wheat than others, both within and between treatments.  This is likely an important cause of variation in disease levels within and between treatments.  We have mentioned this aspect in the Discussion of the revised manuscript which now includes the following passage: “Furthermore, we noted that wheat plants in some plots appeared more vigorously growing than in other plots, which increases WSR severity. The control plots appeared to have the least vigorous plants of all treatments and the other treatment plots varied as well, mostly towards more verdant wheat. While we are not fully certain, the differences in wheat vigor between plots (Fig S3) likely generated the wide variation within and between replicate treatments in the field experiment.”

While not specifically mentioned (but the cited references detail the following methods), during the inoculation process we apply spores in closed-topped chambers which shield the area from wind, wet the foliage down prior to inoculation so that the spores adhere to the plants, inoculate while there are no winds, and then immediately cover the inoculated area with black plastic which protects the plants and inoculum from freezing temperatures and wind dispersal.  It is possible that something unforeseen and undetected occurred over a large number of the plots, but we did not see evidence of such inoculation breaches such as a plume of disease outside of the inoculated area prior to the spray treatments (which we would expect if spores had escaped the inoculation area). 

Reviewer 2 Report

Dear Authors,

The manuscript is really interesting. I have no additional comments. 

Author Response

The manuscript is really interesting. I have no additional comments. In my opinion it can be accepted. 

We did not make any additional comments or edits as reviewer #2 did not suggest any.

Reviewer 3 Report

Wheat stripe rust is air-borne and the spread of this diesease can be affected by direction and intensity of wind. A stripe rust-infected focus is main source of the disease in a larger scale epidemics. Low or high temperatures can extend duration of latent period than optimal temperatures.  Theortically, the less latent period of  the disease, the faster it diffuse to cause outbreak.  In addition, importantly, wheat striep rust is capable for spore reproduction. Single spore can increase 2 million new spores during a 10-day generation. This is main reason of the disese epidemics.  See comments in revised PDF version manuscript. 

Author Response

Reviewer #3

Wheat stripe rust is air-borne and the spread of this diesease can be affected by direction and intensity of wind. A stripe rust-infected focus is main source of the disease in a larger scale epidemics. Low or high temperatures can extend duration of latent period than optimal temperatures.  Theortically, the less latent period of  the disease, the faster it diffuse to cause outbreak.  In addition, importantly, wheat striep rust is capable for spore reproduction. Single spore can increase 2 million new spores during a 10-day generation. This is main reason of the disese epidemics.  See comments in revised PDF version manuscript. 

Reviewer #3 had very helpful editorial and formatting suggestions which we accepted and included in the revision, including the formatting of references in the literature cited section, in text references to the figures, and a few other typos.  Reviewer #3 also had comments attached to the pdf as notes.  We have addressed these comments in the revised version and present the comments below.

Line 128 living host plants are necessary for infection of WSR, but not only be healthy ones. Sometimes, weak or diseased plants by other pathogens can also be infected by this rust.

Response: We omitted healthy and replaced the adjective with "susceptible".

Line 142 it means immune to WSR pathogen races that were used. Thus, change 100% resistance to immune. if not, highly resistant should be suggested to use.

Response: we changed this term to "complete resistance".

Line 144: what is averaged temperatures during the coldest months (last December and next January) in this experiment fields. 

Response: We added the following information to the revised manuscript: "temperatures (mean high temp Dec. = 5.5 C, mean low temp Dec = -3.4 C)."

Line 144: WSR overwinters in this region by slowly-developing urediniospores or hyphae in infected tissue (such as leaf sheath).

Response: We revised the passage to indicate the multiple factors that contribute to a low incidence of naturally occurring wheat stripe rust with the following: "Field experiments took place in a small, geographically isolated arid growing region of central Oregon where the low winter temperatures (mean high temp Dec. = 5.5 C, mean low temp Dec = -3.4 C) are unfavorable for WSR overwintering, together limiting the incidence of non-experimental outbreaks."

Line 183: what kind of inoculation method was used to increase spores or what method was referenced for use. 

Response: We added refernces 29,41 for papers which describe the spore culturing process in detail.

Line 186: Generally, spore  of this rust can germinate at optimal temperatures ranging from 9-13 Celsius and also can germinate above freezing. Under such a low temperatures, how to determine spore germination and  penetration into wheat. 

Response: We modified the revision to indicate that covering the inoculated area with plastic overnight shelters the plants from frost. 

Although we did not include this point in the revision, we know that the inoculations were successful as WSR was observed in all of our inoculated areas and only in our inoculated areas prior to spray treatments.

Round 2

Reviewer 1 Report

I fully acknowledge the effort that authors have invested in this study. I'm overall happy with the responses and the conclusions are generally well balanced with the uncertainties and assumptions of experiment and simulations. 

Latent period: regarding the latent period used for experiment and simulation. I can't find the latent period assumed for the experiment in the text. But it is worth confirming it is indicated there and that it matches the latent period used in the simulation for cross-validation. The response by the authors still didn't response this request. Anyway, for the 16 days used as latent period for the simulation, I'm not against that reasonable value but I think it should be the same than the latent period calculated in the experiment to be fully equivalent and cross-compare. In fact, I seriously doubt that the latent period won't affect the efficacy of early treatments as the authors state in L272. Shorter latent periods due to more favourable weather conditions might shorten the window of opportunity for treatment. In fact the reference provided to support the lack of relevance of the latent period states that latent period has a strong influence in the velocity of the epidemic (see Fig 4 in Sackett, K.E. & Mundt, C.C. 2005)

Parameter R0: I already highlighted that the R0=4 used for simulation of the WSR eradication potential is not biologically plausible. The authors in the response indicated that they use this value "to consider what treatment effects might we expect over a broad range of epidemiological conditions outside what is known for wheat stripe rust". However specifically that section 2.7 is related to WSR eradication (even in the title) so that generality is not really expressed in the current text of the manuscript. 

Regarding the last concern about the inconsistency of results in the experiment for ring cull size, I acknowledge that authors have included another potential explanation that couldn't be controlled. Based on this uncertainty we can't say by the experiment that treatment area was not important. The abstract is conservative in this respect and do not mention this result, leaving just for the simulations any inference about treatment area. 

Minor comments: 

NOT RESOLVED my previous comment in old version L193: this isn’t very clear. What do you mean with “fields blocked”. From the previous paragraph it seems all plots are surrounded by >30 m of non-susceptible cultivar 

L224: The model specifications still require improvement. Remove "generalised" if you use a normal distribution. For transparency please indicate that you use "blocks" as random effect in the model. 

L249: although indicated in the response, the value for parameter “a” is not included in the reviewed text 

Author Response

I fully acknowledge the effort that authors have invested in this study. I'm overall happy with the responses and the conclusions are generally well balanced with the uncertainties and assumptions of experiment and simulations. 

Latent period: regarding the latent period used for experiment and simulation. I can't find the latent period assumed for the experiment in the text. But it is worth confirming it is indicated there and that it matches the latent period used in the simulation for cross-validation. The response by the authors still didn't response this request. Anyway, for the 16 days used as latent period for the simulation, I'm not against that reasonable value but I think it should be the same than the latent period calculated in the experiment to be fully equivalent and cross-compare. In fact, I seriously doubt that the latent period won't affect the efficacy of early treatments as the authors state in L272. Shorter latent periods due to more favourable weather conditions might shorten the window of opportunity for treatment. In fact the reference provided to support the lack of relevance of the latent period states that latent period has a strong influence in the velocity of the epidemic (see Fig 4 in Sackett, K.E. & Mundt, C.C. 2005).

We have attempted to clear up any issues with the latent period by calling attention in the manuscript that we know the temperatures fluctuated enough within the course of the study to deviate from the 16 day latent period.  However, our field observations and growing degree days models suggested only a +1/-1 day potential difference from the 16 days.  In reference to Fig. 4, we do not disagree that the disease abundance values would be different if the latent period were to vary to either 15 or 17 days.  In our study we were not interested in the velocity of the disease isopath (the subject of the referenced paper and Figure 4), but rather if disease escapes treatment and which treatments, timing versus area have the proportionally greatest suppressive effect. Furthermore, the differences in latent period for the referenced Figure was 5 days, not one day.  Based on what was presented in Figure 4, a +/- 1 day would potentially give rise to faster or slower spread of the 20% disease isopath, but the difference would not appear to be substantial based on the referenced Fig. 4. We maintain that qualitatively (as stated originally), our interpretation of the time versus area treatment effects is not likely to impacted by the plus or minus one day latent period deviation in the field study or the simulations. 

The modified passage in the revision (lines 282-289) was an attempt to clarify any confusion regarding the proposed and estimated (experienced) latent period from the field and in silico studies.  The new passage now reads: “We used a latent period and infectious period of 16 days based on climate conditions, degree-day models, and field observations gathered over the duration of the field study. Although the WSR latent period may vary with temperature [53] and the field conditions did fluctuate with some temperatures being warmer or colder depending on the week, the +/- 1 day weekly variation from the overall 16 day latent period caused by weather in the field experiment is unlikely to substantially influence our simulation results in a manner that would qualitatively alter our interpretation of the relative influence of timing and area treatment effects [57].”    

Parameter R0: I already highlighted that the R0=4 used for simulation of the WSR eradication potential is not biologically plausible. The authors in the response indicated that they use this value "to consider what treatment effects might we expect over a broad range of epidemiological conditions outside what is known for wheat stripe rust". However specifically that section 2.7 is related to WSR eradication (even in the title) so that generality is not really expressed in the current text of the manuscript. 

We modified the revised version (see lines 399-403) to make clear that the low reproductive number is an extreme and perhaps unlikely WSR scenario.  The sentences now read: “Consequently, we evaluated the potential for eradication at only the earliest timing (1.125 LP) and with an R0 value of 4 (an extreme WSR scenario), as it was obvious that an epidemic with a greater R0 would not be contained by our treatments. A R0 value of 4, could, for example, represent an uncommon, hypothetical biological situation where a mostly WSR resistant, but not entirely resistant, wheat cultivar is grown.”

Regarding the last concern about the inconsistency of results in the experiment for ring cull size, I acknowledge that authors have included another potential explanation that couldn't be controlled. Based on this uncertainty we can't say by the experiment that treatment area was not important. The abstract is conservative in this respect and do not mention this result, leaving just for the simulations any inference about treatment area. 

Minor comments: 

NOT RESOLVED my previous comment in old version L193: this isn’t very clear. What do you mean with “fields blocked”. From the previous paragraph it seems all plots are surrounded by >30 m of non-susceptible cultivar 

We have added the following more detailed description to the manuscript lines 195-204 (revised version): “We used a plot arrangement design within each of four field regions (blocks which contained a replicate of all treatments), such that we could evaluate potential field region effects that was due to an unforeseen environmental gradient if viewed from a north-south perspective or a east-west perspective. This nested random arrangement of experimental plots was used to evaluate the potential interactions of environmental conditions (e.g. soil richness, irrigation patterns, slope, fertilizer, drought stress, etc.) within specific regions of the field. The capacity to statistically evaluate region effects due to east-west and north-south arrangements was the most thorough way to account for larger scale spatial differences that we could not have foreseen during the fall planting and spring growing seasons.”  While the term “sudoku” is not a experimental design, this is in practice how blocks, plots, and treatment replicates were arranged within a field. We hope the added three sentences is a sufficient description.

L224: The model specifications still require improvement. Remove "generalised" if you use a normal distribution. For transparency please indicate that you use "blocks" as random effect in the model. 

Thank you, these recommended changes were made to the manuscript and now appear in the methods section lines 224-225.

L249: although indicated in the response, the value for parameter “a” is not included in the reviewed text 

We apologize for this mistake and thank you for catching this.  It was our mix up on the revised version that was submitted at the time we responded to the comments, the current revision has “a = 428” on line 260.